# Modeling of Accumulator in Roll-to-Roll Coating Equipment and Tension Control with Nonlinear PID

**DOI:** 10.3390/polym16243479

**Published:** 2024-12-13

**Authors:** Guoli Ju, Shanhui Liu, Lei Feng, Chaoyue Wang, Kailin Yang

**Affiliations:** 1Faculty of Printing, Packaging Engineering and Digital Media Technology, Xi’an University of Technology, Xi’an 710048, China; 2220821107@stu.xaut.edu.cn (G.J.); 2220821141@stu.xaut.edu.cn (C.W.); YKL0818@outlook.com (K.Y.); 2Shaanxi Beiren Printing Machinery Co., Ltd., Weinan 714000, China; daleilei1986@126.com

**Keywords:** polymer coating material, roll-to-roll, accumulator, nonlinear PID, tension control

## Abstract

This paper addresses the issue of the high-precision control of substrate tension in an accumulator during the roll-to-roll coating process. First, a coupling model for tension errors in the substrate within the accumulator is established, along with dynamic models for the input–output rollers, carriage, and the thrust model of the ball screw. Based on these models, a simulation model is built in MATLAB/Simulink to analyze the main causes of substrate tension errors in the accumulator under uncontrolled conditions. Next, to tackle the tension errors caused by carriage displacement, a nonlinear proportional–integral–derivative (PID) controller is proposed, and a control strategy for substrate tension in the accumulator is designed. Finally, based on the established simulation model, experiments are conducted using the proposed nonlinear PID controller and the designed tension control strategy, and their performance is compared with that of a classical PID controller. The simulation results show that both the nonlinear PID controller and the classical PID controller, when combined with the proposed tension error control strategy, can reduce tension errors in the accumulator substrate. However, the nonlinear PID controller is more suitable for controlling substrate tension errors in the accumulator. On the one hand, the nonlinear PID controller has better anti-disturbance capability. In the anti-disturbance experiment, under PID control, the substrate tension error remains stable at around −1.6 N, with tension disturbances of ±0.2 N occurring at approximately 185 s and 135 s. On the other hand, the nonlinear PID controller demonstrates better robustness. In the robustness experiment, under the nonlinear PID controller, the substrate tension error fluctuates within the range of 0 to 0.02 N, showing excellent robustness.

## 1. Introduction

Coating slurry or molten polymers onto various film substrates enables the formation of composites that meet specific environmental requirements; this approach is commonly used in the modification of paper and plastic film materials, and many scholars have conducted extensive research in this area. Pieters et al. developed stable aqueous dispersions of poly(3-hydroxybutyrate-co-3-hydroxyvalerate) and successfully used them as a barrier coating material on paper. This significantly enhanced the paper’s wet mechanical properties, water vapor barrier, oil resistance, and water absorbency [1]. Gao et al. successfully synthesized lignin-based polyurethane and applied it to cellulose paper, resulting in a degradable mulch film with outstanding water resistance, effective water vapor barrier properties, and excellent thermal insulation performance [2]. Kunam et al. developed a water-resistant paper using natural rubber latex from Hevea brasiliensis and a butyl stearate hydrophobic coating material; this innovation significantly enhances the paper’s hydrophobicity, water resistance, and mechanical strength [3]. Inthamat et al. developed an active chitosan-based coating material enriched with natural antioxidants and crosslinkers like astaxanthin and genipin, which significantly improved the functional properties of Kraft paper [4]. Wang et al. successfully developed food-safe greaseproof paper by applying a coating material of chitosan and montmorillonite; this innovative coating material significantly improved the paper’s oil resistance and mechanical strength while reducing air permeability [5]. Calosi et al. developed high-solid polylactic acid aqueous dispersions using PEG–PLA–PEG block copolymer as a surfactant, applying them as a hydrophobic coating material on paper; this innovative coating significantly improves the paper’s barrier properties against liquids [6]. Meng et al. created an all-organic PTFE-coated PVDF composite film that features a low conduction loss and a high breakdown strength, making it ideal for energy storage applications [7]. Luca Panariello et al. developed an environmentally friendly food packaging material with antimicrobial properties and enhanced gas barrier performance by applying a coating material made from crustacean and fungal chitin to biodegradable bioplastic films [8].

At present, the ideal equipment for applying these modified polymer coating materials to paper or plastic film is roll-to-roll coating equipment, which has continuous, efficient, and green manufacturing characteristics and is recognized in industry as the ideal equipment for achieving large-scale coating manufacturing; additionlaly, it has a wide range of applications in the field of coating manufacturing. Kim et al. enhanced the thermal stability of polyethylene lithium-ion battery separators by coating a polymer through a roll-to-roll gravure coating process [9]. Schiessl et al. examined the effects of slot–die and reverse gravure coating techniques in roll-to-roll processing for applying a silicate–polyvinyl alcohol composite barrier lacquer. The study focused on how solid content, coating geometry, and the coating gap influence oxygen barrier performance [10]. Choi et al. explored ceramic-coated separators (CCSs) for lithium-ion batteries and successfully produced ultra-thin CCS rolls continuously through a pilot-scale direct-current roll-to-roll sputtering process. Their results indicate that this roll-to-roll method facilitates the large-scale, high-speed production of ultra-thin CCS rolls [11]. Jung et al. explored the use of roll-to-roll dual-layer slot die coating technology to manufacture chitin and cellulose–oxygen barrier films for renewable packaging. This dual-layer slot die coating method achieved lower oxygen permeability than traditional spray coating while completing the double-layer application in a single pass [12]. Esfahani et al. successfully developed a scalable aqueous-phase method for fabricating reduced graphene oxide nanofiltration membranes using an integrated roll-to-roll process; this approach showcases the potential for the large-scale, rapid production of high-performance nanofiltration membranes with roll-to-roll equipment [13]. In the roll-to-roll coating manufacturing process, maintaining high-precision tension control across various functional units is essential to ensure the quality of coated products. Many scholars have conducted extensive research in this area, focusing primarily on tension system modeling and tension control strategies. In the field of tension system modeling, Shin proposed a mathematical model for the tension of a moving web that incorporates various factors influencing tension changes, including the material’s Young’s modulus, coefficient of thermal expansion, and thermal strain [14]. Lee et al. expanded Shin’s mathematical model of web tension by incorporating thermal strain resulting from temperature fluctuations during the drying process in roll-to-roll systems [15]. Jabbar et al. modeled the tension in a multi-span reel-to-reel system, considering the thermal effects of oven temperature changes and the influence of the material’s viscoelastic properties on substrate tension [16]. Wang et al. established tension models for the unwinding, printing, and rewinding subsystems and coupled them in series to create a global tension coupling model [17]. In the field of tension control, numerous scholars have proposed various methods, including fuzzy proportional–integral–derivative (PID) [18], fuzzy fractional-order PID [19], feed-forward PID [20], and feed-forward PID with parameter self-tuning [21]. These approaches, built on traditional PID control, have demonstrated significant research on and applications in substrate tension control for roll-to-roll equipment. Additionally, as control technology continues to advance, adaptive dynamic programming (ADRC) [22,23] and improvement-ADRC [24] controllers have also contributed substantially to this area of research.

However, the control of the tension of the substrate in the accumulator on the rewinding side is more difficult than in other functional units of the roll-to-roll equipment because the accumulator is constantly moving up and down during the operation of the accumulator mechanism, the length of the substrate is constantly changing, and there are multiple inputs during the operation of the accumulator mechanism, which will have an effect on the tension of the substrate in the accumulator. David et al. established nonlinear tension coupling models for both pneumatically driven accumulators and electrically driven accumulators [25,26]. Prabhakar R et al., based on the traditional two-roller tension coupling model and under the assumption that the substrate tension in the storage frame is the same, established a tension coupling model in line with the characteristics of the accumulator tension system [27,28]. In terms of tension control methods for the substrate within the accumulator, the classic PID controller is still the main approach. However, with the increasing prevalence of roll-to-roll coating processes, ultra-thin substrates such as polyethylene film, polypropylene film, polyethylene terephthalate film, and polyvinyl chloride film require high-precision tension control. The classic PID controller, due to its structural limitations, can no longer meet the tension control precision required for the coating of these ultra-thin substrates. Therefore, it is necessary to seek a suitable tension controller to improve the tension control precision in the accumulator on the rewinding side of the roll-to-roll coating manufacturing process.

This paper addresses the tension control problem of the accumulator mechanism on the rewinding side in the roll-to-roll coating manufacturing process. In the first part, based on the traditional two-roll tension coupling model, a tension error coupling model for the accumulator on the rewinding side is established, along with the dynamic models of the input/output rollers and the carriage, as well as the thrust model of the ball screw, and a simulation model is built in MALBA/Simulink 2019a version to analyze the main causes of substrate tension errors within the accumulator under without controller conditions. In the second part, to reduce the substrate tension errors in the accumulator, a nonlinear PID controller is proposed, and a control strategy for the substrate tension within the accumulator is designed. The third part utilizes the simulation model established in the first part to conduct simulation experiments using the nonlinear PID controller and the designed tension control strategy from the second part, verifying the effectiveness of the proposed nonlinear PID controller and control strategy in reducing substrate tension errors within the accumulator. The fourth part provides a summary of the entire study.

## 2. Accumulator Tension Error Coupling Modeling and Analysis

### 2.1. Tension Error Coupling Modeling

In the roll-to-roll manufacturing process using various polymer coatings, the material accumulator system on the rewinding side is designed to operate as follows: When the rewinding roller slows down and stops for a roll change, the carriage moves upward to store the substrate within the system. Once the roll change is completed and the rewinding roller accelerates to its normal speed, the carriage moves downward to release the substrate stored in the system. Throughout the process, the linear speed of the upstream processing section remains constant, and the tension in the substrate within the accumulator system is stabilized within a range. Figure 1a shows the structure of an accumulator system in an industrial site. The accumulator system consists of three main parts: the accumulator, the carriage, and the dancer roller. The accumulator consists of two rows of free rollers, when the rolls on the rewinding side are replaced, the upper row of free rollers moves up and down to store and release the substrate in order to ensure that the speed of the processing section remains unchanged and the continuity of production is maintained. The carriage is connected to the upper row of the free roller, and the motor drives screw the carriage up and down to achieve the upper row of free roller movement and ensure that the accumulator inside the substrate tension stability. The dancer roller is located after the accumulator input roller, and it is responsible for the capture of substrate tension and suppression of substrate tension mutations [29,30]. In the simplified structural diagram of the accumulator system of Figure 1b, the linear speed of the input rollers is *v*_0_, which is the same as the linear speed of the upstream processing section. The linear speed of the output rollers is *v_n_*, matching the linear speed of the rewind rollers. The distance between the upper row of rollers and the lower row of rollers when the carriage is moving is *L*, and the moving speed is *v_c_*.

Under the assumption that there is no slip between the substrate and each free roller, the linear velocities of the first and last free rollers in the accumulator are *v*_0_ and *v_n_*, and the lengths of each substrate segment in the accumulator are *L*. According to the existing tension coupling model of the two rollers [17], the tension coupling model of the substrate in the accumulator is established as follows:(1)$$L\left(t\right)\frac{dT_{i}\left(t\right)}{dt}=\left[\mathrm{AE}-T_{i}\left(t\right)\right]v_{i}\left(t\right)-\left[AE-T_{i-1}\left(t\right)\right]v_{i-1}\left(t\right)+\left[\mathrm{AE}-T_{i}\left(t\right)\right]\frac{dL(t)}{dt}$$

The meanings of the parameters in Equation (Equation 1) are given in Table 1 below:

As shown in Figure 1b, the input and output rollers of the accumulator system are driven by a motor and are equipped with an impression roller, while the other rollers are designated as free rollers. Due to the uniform state of the substrate between all free rollers, the tension in each section can be considered the average tension *T_a_* [27], which is expressed as follows:(2)$$T_{a}=\frac{1}{n}\sum_{i=1}^{i=n}T_{i}$$

Bringing Equation (Equation 2) into Equation (Equation 1) gives the following:(3)$$\left\{\begin{array}{c} \frac{dT_{a}\left(t\right)}{dt}=\frac{1}{n}\frac{\mathrm{AE}}{L(t)}\left[v_{n}\left(t\right)-v_{0}\left(t\right)\right]+\frac{1}{n}\frac{1}{L(t)}\left[T_{0}v_{0}\left(t\right)-T_{n}v_{n}\left(t\right)\right]+\frac{\mathrm{AE}}{L(t)}v_{c}\left(t\right)+\frac{T_{a}\left(t\right)}{L(t)}v_{c}\left(t\right)\\ L^{\prime }\left(t\right)=v_{c}\left(t\right) \end{array}\right.$$

Equation (Equation 3) expresses the relationship between substrate tension and *v*_0_, *v_n_*, and *v_c_* within the accumulator since the second and fourth terms on the right-hand side are far less than the product of *A*, *E* [27], which can accordingly be ignored. If *T_a_* is expressed as the sum of the tension error *e*(*t*) and the reference tension *T_r_*, then Equation (Equation 3) is converted to the following:(4)$$\left\{\begin{array}{c} \frac{de(t)}{dt}=\frac{1}{n}\frac{\mathrm{AE}}{L(t)}\left[v_{n}\left(t\right)-v_{0}\left(t\right)\right]+\frac{\mathrm{AE}}{L(t)}v_{c}\left(t\right)\\ L^{\prime }\left(t\right)=v_{c}\left(t\right) \end{array}\right.$$

Equation (Equation 4) expresses the relationship between the *v*_0_, *v_n_*, and *v_c_* with *e*(*t*) for the tension error coupling model of the accumulator, and according to the literature [27], the dynamics model of *v*_0_, *v_n_*, and *v_c_* is introduced as follows:(5)$$\left\{\begin{array}{c} v_{0}^{\prime }\left(t\right)=\frac{1}{J}\left[-f_{m0}v_{0}\left(t\right)+R^{2}e\left(t\right)+J_{M0}\frac{R}{R_{M0}}u_{M0}+R^{2}\delta_{1}\right]\\ v_{n}^{\prime }\left(t\right)=\frac{1}{J}\left[-f_{mn}v_{n}\left(t\right)-R^{2}e\left(t\right)+J_{Mn}\frac{R}{R_{Mn}}u_{Mn}+R^{2}\delta_{2}\right]\\ v_{c}^{\prime }\left(t\right)=\frac{1}{M_{c}}\left[u_{c}-ne\left(t\right)-nT_{r}+\delta_{3}\right]-\left(1+\mu \right)g \end{array}\right.$$

The meaning of the parameters in Equation (Equation 5) is shown in Table 2:

Equation (Equation 5) states that the input roller line speed *v*_0_ is the same as the upstream processing section line speed. The output roller line speed *v_n_* follows the changes in rewinding roller line speed, as shown in the speed change curve in Figure 2a. The carriage moving speed *v_c_* follows the changes in *v*_0_ and *v_n_*. Equation (Equation 6) provides the specific calculation equation, and the speed curve is shown in Figure 2b.
(6)$$v_{c}\left(t\right)=\frac{v_{0}\left(t\right)-v_{n}\left(t\right)}{n}$$

In Figure 2, the variation of *v_n_* is split into eight phases, with the corresponding change in *v_c_* also divided into eight phases when the rewinding side rollers are replaced. The operation of the carriage in each phase is illustrated in Figure 3.

The control quantity *u_c_* mentioned in Equation (Equation 5) refers to the thrust force *F_s_*(*t*) generated via the ball screw. As illustrated in Figure 4, the motor with the reducer rotates the coupling, which in turn interacts with the ball screws located at both ends of the carriage to create a combined force *F_s_*(*t*). This combined force eventually propels the carriage to move upwards or downwards. In Figure 4, $\tau_{1}$, $\tau_{2}$, and $\tau_{3}$ represent the output torque of the motor, coupling, and ball screw, respectively; since the ball screws at both ends rotate together because of the coupling, the final thrust force formed via the motor in the whole process is *f*(*t*) = *F_s_*(*t*)/2.

According to the torque transfer relationship illustrated in Figure 4, the relationship between $\tau_{1}$ and $\tau_{3}$ can be expressed as shown in Equation (Equation 7):(7)$$\tau_{1}=\eta_{1}\eta_{2}N_{1}N_{2}\tau_{3}$$

Incorporating the ball screw thrust is $f\left(t\right)=2\pi \eta_{3}\tau_{3}/L_{s}$ and motor torque is $\tau_{1}=J_{Ms}v_{Ms}/R_{Ms}$, Equation (Equation 7) can be converted to the following:(8)$$f\left(t\right)=\frac{2J_{Ms}\pi \eta_{3}}{\eta_{1}\eta_{2}N_{1}N_{2}L_{s}R_{Ms}}v_{Ms}$$

Equation (Equation 8) expresses the relationship between motor speed *v_Ms_* and ball screw thrust *f*(*t*), and the meanings of the parameters in Equation (Equation 8) are shown in Table 3:

### 2.2. Coupled Model Simulation and Analysis

To analyze the main causes of tension errors in the substrate within the accumulator, this section describes the utilization of the established tension error coupling model of the accumulator to build a simulation model in MATLAB/Simulink 2019a version and conduct simulation experiments without using any controllers.

In the simulation, a polyethylene terephthalate (PET) film at 20 °C was selected as the substrate. The cross-sectional area *A* and elastic modulus *E* of the film are provided in Table 4. The input roller speed *v*_0_ of the accumulator is set to two operational speeds, 1 m/s and 3 m/s, allowing for the observation of the tension errors in the PET film within the accumulator at these different speeds. When the input roller speed *v*_0_ of the accumulator is set to 1 m/s and 3 m/s, the ideal speed curve *v_n_ref_* for the output roller is planned based on the actual operating conditions of the output roller. Additionally, the ideal speed curve *v_c_ref_* for the carriage is determined according to Equation (Equation 6). The curves *v_n_ref_*, *v_c_ref_* are illustrated in Figure 5.

Based on the specific simulation parameters in Table 4 and the ideal speed profile in Figure 5, a simulation model was developed in MATLAB/Simulink using Equations (4), (5) and (7). The simulation step size was set to 1 s, with a total run time of 200 s. To better analyze the impact of the accumulator carriage on substrate tension, the values of δ_1_, δ_2_, and δ_3_ in Equation (Equation 5) were set to 0 N. The results of the substrate tension error *e* and the displacement error *e_xc_* of the accumulator carriage during the simulation are shown in Figure 6.

From Figure 6a,b, it is clearly observed that, without a controller, when the accumulator input roller speed *v*_0_ is set to 1 m/s and 3 m/s, significant tension errors and fluctuations occur in the substrate. As the speed increases, both the magnitude and range of these tension errors also increase. At 1 m/s, the maximum fluctuation of tension can reach approximately 36 N, while at 3 m/s, it increases to about 135 N. It can also be observed that, although the speed errors between the accumulator input speed *v*_0_ and the output roller speed *v_n_* are nearly zero, the displacement error of the carriage is relatively large, with fluctuations around 0.28 m. These displacement errors fluctuate in tandem with the tension errors, and as *v*_0_ increases, both the displacement error and its fluctuation range are further amplified.

Therefore, it can be inferred that the main reason for the tension errors and fluctuations of the substrate within the accumulator is that the running speed *v_c_* of the accumulator carriage cannot effectively track the ideal speed curve *v_c_ref_* during certain periods. This leads to displacement errors and fluctuations of the carriage, resulting in tension errors and fluctuations of the substrate within the accumulator during those phases. In other words, when the running speed *v_c_* of the accumulator carriage can effectively track the ideal speed curve *v_c_ref_* and there are no displacement errors, the tension error of the substrate within the accumulator is 0 N.

In order to verify the conclusions drawn from Figure 6, the simulation experiment is set up in the ideal situation, when the accumulator carriage running speed *v_c_* is able to track the ideal speed curve *v_c_ref_* completely, and there is no carriage displacement error, the substrate tension error in the accumulator is observed. The final simulation results show that the substrate tension error at no controller, in the ideal state with 1 m/s and 3 m/s simulation speeds, is 0 N, which is in accordance with the above conclusion.

In summary, based on the established tension error coupling model for the accumulator, the input–output roller and carriage dynamics models, as well as the ball screw thrust model, a simulation model was developed in MATLAB/Simulink 2019a version. Under the condition of no controller, the analysis identified the main source of substrate tension error within the accumulator. These errors mainly arise because the running speed *v_c_* of the accumulator carriage fails to adequately track the ideal speed curve *v_c_ref_*, resulting in displacement errors in the carriage. Ultimately, this leads to tension errors and fluctuations in the substrate. Therefore, addressing this issue is essential for effectively controlling tension errors within the accumulator.

## 3. Controller Design and Analysis

This section employs a nonlinear PID controller to tackle the issue of the accumulator carriage speed *v_c_* inadequately tracking the ideal speed curve *v_c_ref_*, resulting in tension errors and fluctuations in the substrate. The section begins with an introduction to the main components of the controller, followed by the presentation of the overall control strategy for the tension system of the accumulator mechanism. Finally, simulation experiments will be conducted based on the established model to validate the overall effectiveness of the proposed nonlinear PID controller.

### 3.1. Controller Design

The classic PID controller generates the control signal *u* for the controlled object based on the error *e* between the target setpoint *v* and the system output *y*, along with the integral and derivative of the error, using a linear combination [31]. This method is widely applied in various industrial fields. However, the classic PID controller involves limitations, such as its inability to effectively track abrupt changes in external signals and the fact that the linear combination may not always yield the most efficient control [31]. As a result, its performance may not be satisfactory in certain applications. In this paper, on the one hand, the ideal speed curve *v_c_ref_* for the accumulator carriage has many sudden change points; on the other hand, there is a high precision requirement for the accumulator carriage to track the ideal speed curve *v_c_ref_*. Therefore, it is necessary to seek a controller that can solve the issue of the accumulator carriage’s running speed *v_c_* not being able to accurately track the ideal speed curve *v_c_ref_* at certain times.

The scholar Han introduced the characteristics of and construction method for the nonlinear PID controller in [32]. Based on this construction method, this paper proposes a nonlinear PID controller. The nonlinear PID controller used in this study first employs two second-order tracking differentiators, TD_1_ and TD_2_, to track the input and output signals, respectively, and provide the differentiated signals of the input and output. Meanwhile, the control signal *u* is derived using the nonlinear error state feedback (NLSEF) method. This control approach is characterized by the stable tracking of the input and output signals, the effective extraction of the differentiated signals, and the efficient generation of the control output. These features effectively address some of the limitations of the classical PID controllers. The specific structure of the nonlinear PID controller is as follows.

TD1:(9)$$\left\{\begin{array}{c} e_{TD1}\left(k\right)=x_{1}\left(k\right)-y_{ref}\left(k\right)\\ fh\left(k\right)=\mathrm{fhan}\left(e_{TD1}\left(k\right),x_{2}\left(k\right),r_{0},h_{0}\right)\\ x_{1}\left(k+1\right)=x_{1}\left(k\right)+hx_{2}\left(k\right)\\ x_{2}\left(k+1\right)=x_{2}\left(k\right)+hfh\left(k\right) \end{array}\right.$$

TD2:(10)$$\left\{\begin{array}{c} e_{TD2}\left(k\right)=z_{1}\left(k\right)-y\left(k\right)\\ fh\left(k\right)=\mathrm{fhan}\left(e_{TD2}\left(k\right),z_{2}\left(k\right),r_{1},h_{1}\right)\\ z_{1}\left(k+1\right)=z_{1}\left(k\right)+hz_{2}\left(k\right)\\ z_{2}\left(k+1\right)=z_{2}\left(k\right)+hfh\left(k\right) \end{array}\right.$$

NLSEF:(11)$$\left\{\begin{array}{c} e_{1}\left(k\right)=x_{1}\left(k\right)-z_{1}\left(k\right)\\ e_{2}\left(k\right)=x_{2}\left(k\right)-z_{2}\left(k\right)\\ u\left(k\right)=-\;\mathrm{fhan}\;\left[e_{1}\left(k\right),ce_{2}\left(k\right),r_{2},h_{2}\right] \end{array}\right.$$

The *fhan* function in Equations (9)–(11) above is as follows:(12)$$\left\{\begin{array}{c} d=rh,\;d_{1}=hd\\ y=x_{1}+hx_{2}\\ g\left(x\right)=\left\{\begin{array}{cc} x_{2}-\mathrm{sign}\left(y\right)\frac{\left(d-\sqrt{d^{2}+8r\left|y\right|}\right)}{2}, & |y|>d_{1}\\ x_{2}+y/h, & \left|y\right|\leq d_{1} \end{array}\right.\\ \;\mathrm{fhan}\;\left(x_{1},x_{2},r,h\right)=\left\{\begin{array}{cc} -r\mathrm{sign}[g(x)], & |a|>d\\ -r\frac{g(x)}{d}, & |a|\leq d \end{array}\right. \end{array}\right.$$

In Equation (Equation 9), *x*_1_ represents the quantity tracking the reference input *y_ref_*, *x*_2_ represents the differentiated signal of the reference input *y_ref_*, and *h* is the integration step size. In Equation (Equation 10), *z*_1_ represents the quantity tracking the system output *y*, *z*_2_ represents the differentiated signal of the system output *y*, and *h* is the integration step size. In Equation (Equation 11), *e*_1_ is the difference between *x*_2_ and *z*_1_, which can be considered the system error, while *e*_2_ is the difference between *x*_2_ and *z*_1_ in terms of their derivatives, which can be regarded as the differential of the system error. *u* is the system control output obtained using the *fhan* function, where *c* in the function is called the damping factor. The *fhan* function in Equation (Equation 12) is referred to as the fastest control synthesis function. It can suppress the chattering phenomenon. The parameters *r*_0_, *r*_1_, and *r*_2_ determine the response speed and are called the speed factors. The parameters *h*_0_, *h*_1_, and *h*_2_ determine the filtering effect and are called the filtering factors. To better suppress noise, *h*_0_, *h*_1_, and *h*_2_ are typically set to be larger than *h* [33,34].

Figure 7a illustrates the control strategy of the simulation model built in Simulink. Initially, three PID controllers are used to control the input–output rollers and the carrier of the accumulator, with the three PID controllers tracking the ideal speed profiles *v_n_ref_*, *v*_0_*ref*_, and *v_c_ref_*. Then, a nonlinear PID controller is set up to control the tension error of the substrates within the accumulator. Specifically, TD_1_ is responsible for tracking the tension error reference input *e_ref_* (*t*), where *y_ref_* in Equation (Equation 10) represents the input *e_ref_* (*t*). TD_2_ tracks the tension error feedback *e*(*t*), with *y* in Equation (Equation 11) representing the feedback value *e*(*t*). The NLSEF is responsible for providing the adjustment value Δ*v_c_* to the carriage’s running speed, where *u* in Equation (Equation 12) represents the adjustment value Δ*v_c_*. Figure 7b provides an overview of the tension control strategy for the substrate within the accumulator in an industrial setting. Three motor drivers regulate *v_n_*, *v*_0_, and *v_c_* to align with *v_n_ref_*, *v*_0_*ref*_, and *v_c_ref_*, respectively. Additionally, a separate controller computes the adjustment value Δ*v_c_* for the carriage’s running speed based on the deflection angle from the dancer roller and relays this information to the motor driver controlling the carriage’s movement.

### 3.2. Simulation and Analysis

In this section, based on the simulation model developed in the coupled model simulation analysis in Section 2, the effectiveness of the tension control strategy shown in Figure 7, as well as the anti-interference ability and robustness of the nonlinear PID controller, will be simulated and analyzed under conditions where the accumulator system possesses internal disturbances, varying operating speeds, different coating substrates, and different experimental temperatures. The results will be compared with the tension control effect of the classical PID controller. The simulation experiment is divided into two parts, as follows.

#### 3.2.1. Anti-Interference Experiments

The anti-interference experiments were based on the simulation model established in the coupled model simulation analysis in Section 2. Two sets of experiments were set up using the simulation parameters from Table 4 and the speed profiles shown in Figure 5; the details are as follows.

Part 1: Set *v*_0_*ref*_ as 1 m/s and 3 m/s. The reference curves for *v_n_*__*ref*_ and *v_c_ref_* are displayed in Figure 5a and Figure 6b. Using $\delta_{1}$, $\delta_{2}$, and $\delta_{3}$ in Equation (Equation 5) as 20 N, the nonlinear PID controller is applied for tension error control. The controllers involved in the simulation process are shown in Table 5. The tension error *e* of the substrate in the accumulator and the displacement error *e_xc_* of the pallet during the simulation are shown in Figure 8a,b.

Part 2: Set *v*_0_*ref*_ as 1 m/s and 3 m/s, with the reference curves for *v_n_ref_* and *v_c_ref_* displayed in Figure 5a,b. Using $\delta_{1}$, $\delta_{2}$, and $\delta_{3}$ in Equation (Equation 5) as 20 N, the classical PID controller is utilized for tension error control. The controllers involved in the simulation process are shown in Table 5. The tension error *e* of the substrate in the accumulator and the displacement error *e_xc_* of the pallet during the simulation are presented in Figure 8a,b.

Figure 8 illustrates the control performance of the nonlinear PID and PID controllers on the tension error of the substrate in the accumulator and the displacement error of the carriage. Based on the details in Figure 8, the following analysis can be made:

Firstly, under the same simulation conditions, the tension fluctuation of the substrate in Figure 6 can reach a maximum of 36 N and 135 N, respectively, while in Figure 8, under nonlinear PID control, the substrate tension stabilizes around 0.035 N and 0.004 N. Under PID control, the substrate tension stabilizes around −1.6 N. Comparing these results reveals that both the nonlinear PID and PID controllers significantly reduce the displacement error of the carriage and decrease the tension error of the substrate in the accumulator. The simulation results validate the effectiveness of the tension control strategy presented in Figure 8. Additionally, the displacement errors of the carriage in Figure 8 are much smaller than those in Figure 6, and the corresponding tension errors are also considerably lower, which aligns with the conclusions drawn from the coupled model simulation analysis in Section 2.

Next, the simulation results in Figure 8 demonstrate that the nonlinear PID controller can effectively reduce the displacement error of the accumulator and decrease the tension error of the substrate. Without any controller, as shown in Figure 6, the displacement error of the accumulator fluctuates around 0.28 m, while the maximum fluctuation of the tension is 36 N and 135 N. Under the control of the nonlinear PID, Figure 8 shows that the displacement error of the accumulator stabilizes around 0.1257 m and 0.0697 m, while the corresponding substrate tension stabilizes around 0.035 N and 0.004 N.

Finally, the results in Figure 8 indicate that the nonlinear PID controller exhibits better anti-interference performance compared to the standard PID controller. On the one hand, the tension control precision of the PID controller is lower than that of the nonlinear PID controller. When the sudden changes in tension error shown in Figure 8 are ignored, the tension error of the substrate under PID control stabilizes around −1.6 N, which is nearly two orders of magnitude higher compared to the nonlinear PID. On the other hand, the PID controller tends to generate unreasonable control actions when the external input is a jump signal. At a running speed of 1 m/s, the displacement error of the accumulator stabilizes around 0.14 m, but at 185 s, it suddenly rises to 0.6 m and then falls back to 0.55 m. Similarly, at a running speed of 3 m/s, the displacement error suddenly drops to 0.09 m at 135 s and stabilizes. Such sudden changes are difficult to achieve in practical engineering and can cause significant disturbances to the substrate tension. In Figure 8, these sudden changes result in tension disturbances of ±0.2 N for the substrate at 185 s and 135 s, respectively.

#### 3.2.2. Robustness Experiments

To verify the robustness of the proposed controller, this section sets up simulation experiments from three aspects: system operating speed, substrate type, and experimental temperature. The experiments are described as follows:

Part 1: Set *v*_0-*ref*_ to 2 m/s and 4 m/s, with changes in *v_n_ref_* and *v_c_ref_* shown in Figure 9a,b, and the disturbance factors $\delta_{1}$, $\delta_{2}$, and $\delta_{3}$ in Equation (Equation 5) are set to 20 N. The specific simulation parameters are shown in Table 4. Tension error control is performed using both the nonlinear PID and PID controllers, with the specific parameters of the controllers involved in the simulation process shown in Table 5. The tension error *e* in the substrate inside the accumulator and the displacement error *e_xc_* of the carriage are shown in Figure 9c,d.

Part 2: Based on Part 1, simulation experiments are conducted using PET film at 10 °C and 30 °C, with the specific Young’s modulus and substrate cross-sectional area given in Table 6. Other simulation parameters are listed in Table 4. The tension error *e* in the substrate inside the accumulator and the displacement error *e_xc_* of the carriage are shown in Figure 10a–d.

Part 3: Based on Part 1, simulation experiments are conducted using paper and aluminum foil at 20 °C, with the specific Young’s modulus and substrate cross-sectional area given in Table 6. Other simulation parameters are listed in Table 4. The tension error *e* in the substrate inside the accumulator and the displacement error *e_xc_* of the carriage are shown in Figure 11a–d.

Figure 9, Figure 10 and Figure 11 show the tension error *e* of the substrate inside the accumulator and the displacement error *e_xc_* of the carriage under the control of the nonlinear PID controller and the traditional PID controller, with variations in system speed, experimental temperature, and substrate type. From the simulation results, the following observations can be made:

First, under the same speed conditions, when the experimental temperature and substrate change, the nonlinear PID controller exhibits a smaller tension error, which stabilizes within a very narrow range. As shown in Figure 9, Figure 10 and Figure 11, when *v*_0_*ref*_ is set to 2 m/s and 4 m/s, the tension error of the substrate inside the accumulator under the nonlinear PID controller fluctuates between 0 and 0.02 N, while the traditional PID controller maintains the error around 1.6 N.

Next, under the same speed conditions, a higher Young’s modulus leads to a larger tension error. In Figure 10, the Young’s modulus of PET film decreases with an increasing temperature, and correspondingly, the tension error also decreases as the Young’s modulus becomes smaller at the same speed. In Figure 11, paper has the lowest Young’s modulus, while aluminum foil has the highest. As a result, under the same speed, the tension error for paper is the smallest, while the tension error for aluminum foil is the largest.

Based on the above simulation results, it can be concluded that, under different operating speed conditions, varying experimental temperatures, and different substrate types, the nonlinear PID controller demonstrates excellent tension error control performance. The tension error remains stable within a narrow range, showing strong robustness. Furthermore, the substrate’s Young’s modulus also influences the tension error, with smaller Young’s modulus resulting in smaller tension errors.

## 4. Conclusions

In this paper, a simulation model is developed in MATLAB/Simulink 2019a version based on the established coupling model of the tension error in the accumulator on the rewinding side, the dynamic models of the input and output rollers and carriage, and the dynamics model of the roller ball screw. First of all, under the condition of no controller, the main source of the substrate tension error in the accumulator was analyzed; the substrate tension error in the accumulator was mainly due to the running speed *v_c_* of the accumulator carriage being unable to track the ideal speed curve *v_c_ref_* well, which leads to the displacement error of the accumulator carriage and, ultimately, the substrate tension error and fluctuation. Then, for this problem, a nonlinear proportional–integral–derivative (PID) controller was proposed, and the control strategy of the substrate tension in the accumulator was designed. The nonlinear PID controller outputs the adjustment amount of the running speed of the accumulator carriage Δ*v_c_* based on the feedback and input of the substrate tension error. Finally, based on the simulation model, the accumulator carriage running speed *v_c_* was adjusted using the nonlinear PID controller and the PID controller, and the simulation results show the following:A nonlinear PID and a PID controller can both reduce the tension error of the substrate in the accumulator to a large extent, proving the effectiveness of the tension control strategy shown in Figure 6.The nonlinear PID controller demonstrates a better anti-interference capability. In the anti-interference experiment, when the external input is a step signal, the traditional PID controller tends to generate unreasonable control actions. During the simulation, the tension of the substrate exhibits abrupt changes of ±0.2 N at 185 s and 135 s, respectively.The nonlinear PID controller has better robustness. In the robustness experiment, the nonlinear PID controller shows excellent tension error control, with the tension error fluctuating within the range of 0 to 0.02 N, demonstrating strong robustness.

## Figures and Tables

**Figure 1 polymers-16-03479-f001:**
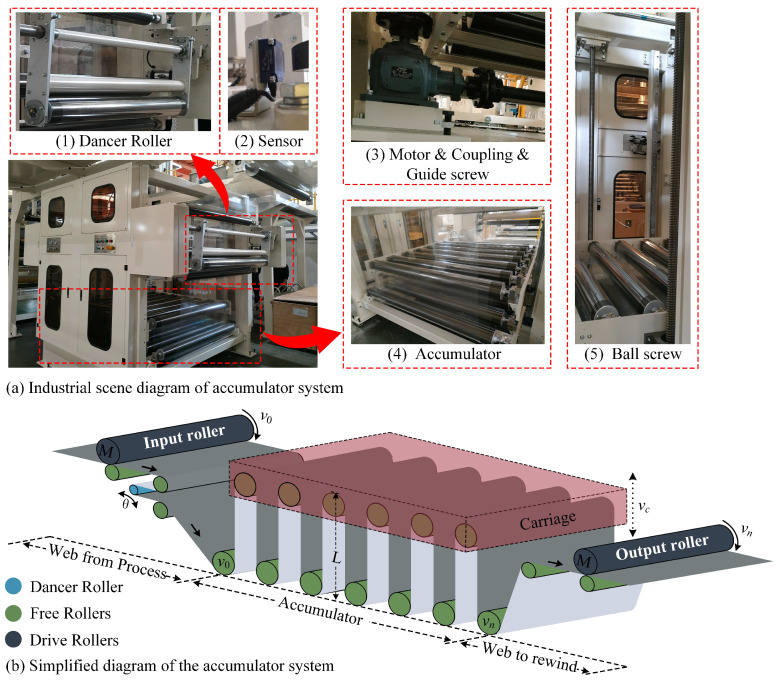
Structural diagram of the accumulator system.

**Figure 2 polymers-16-03479-f002:**
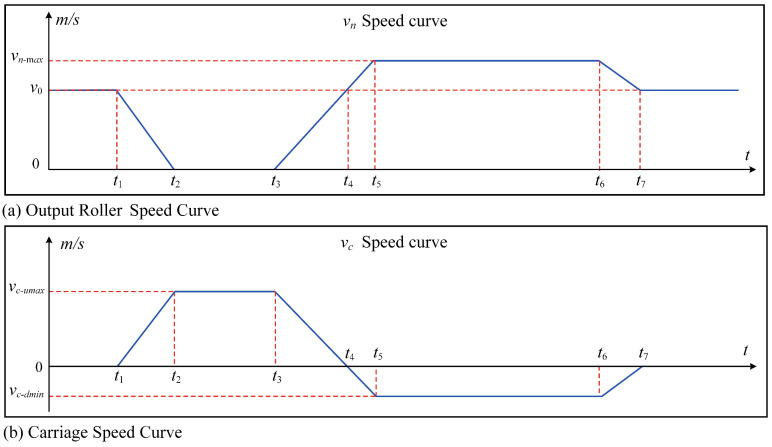
Output roller and carriage speed curves.

**Figure 3 polymers-16-03479-f003:**
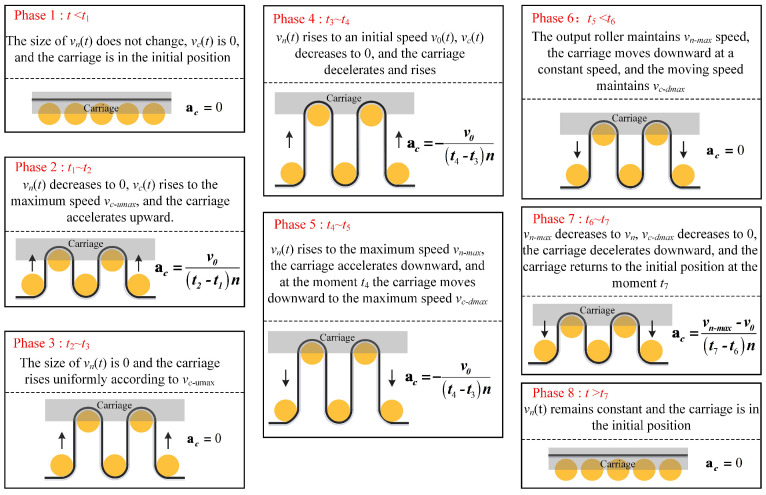
Carriage movement status chart.

**Figure 4 polymers-16-03479-f004:**
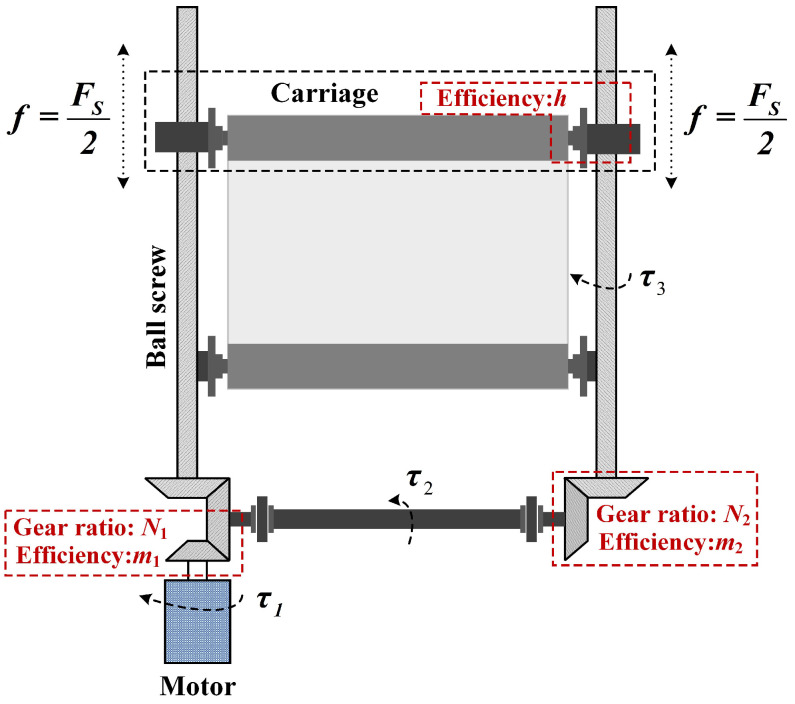
Schematic diagram of carriage structure.

**Figure 5 polymers-16-03479-f005:**
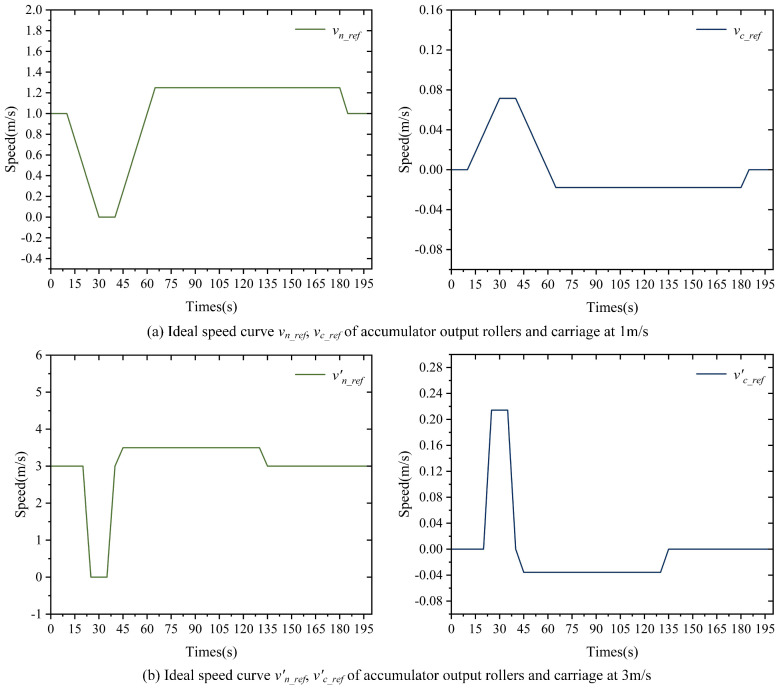
Ideal speed curve.

**Figure 6 polymers-16-03479-f006:**
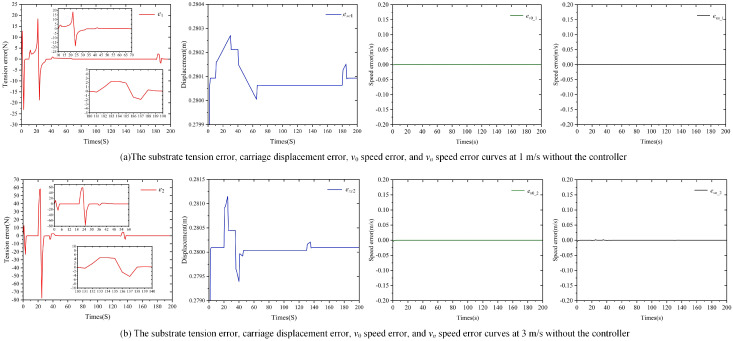
Tension error curve of accumulator and displacement curve of pallet.

**Figure 7 polymers-16-03479-f007:**
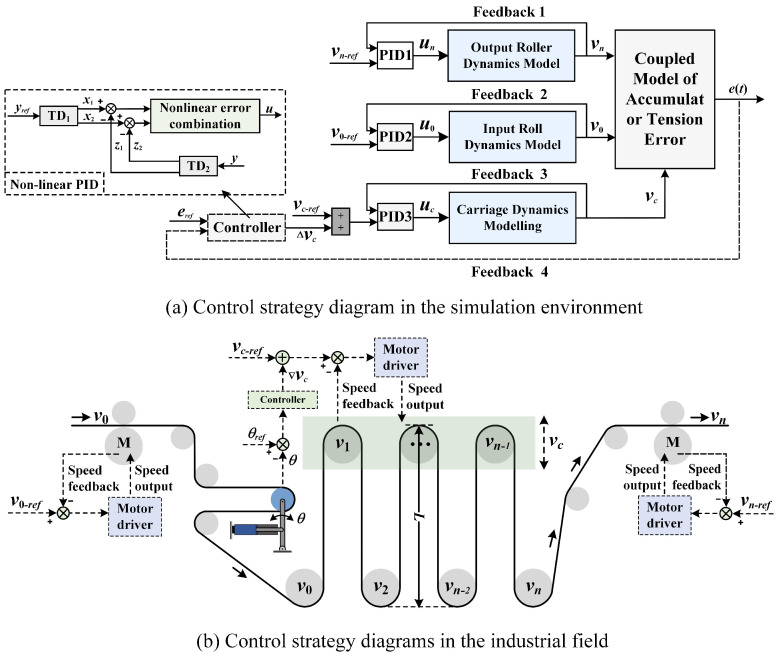
Controller structure diagram.

**Figure 8 polymers-16-03479-f008:**
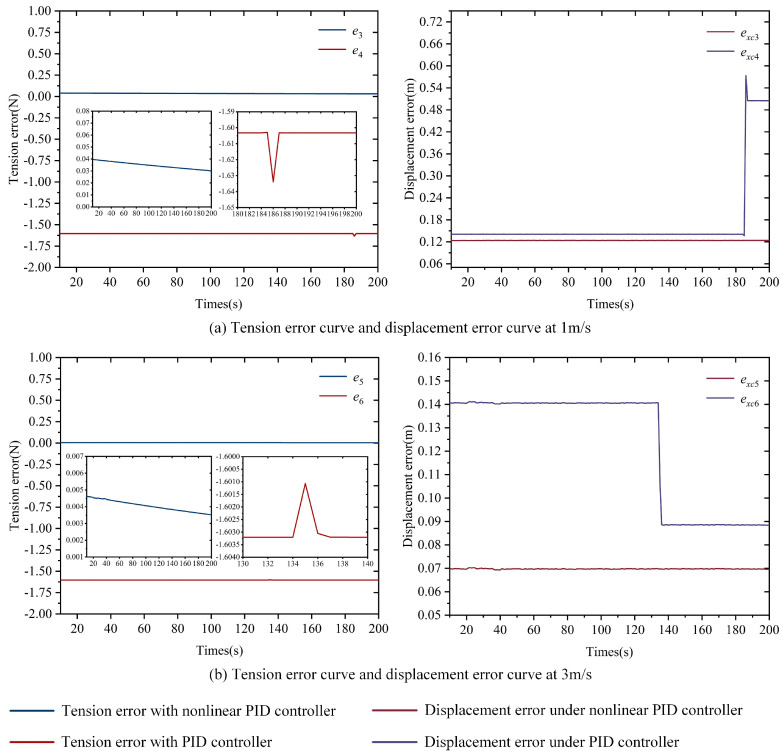
Curve of accumulator substrate tension error and carriage displacement error under nonlinear PID and PID controllers.

**Figure 9 polymers-16-03479-f009:**
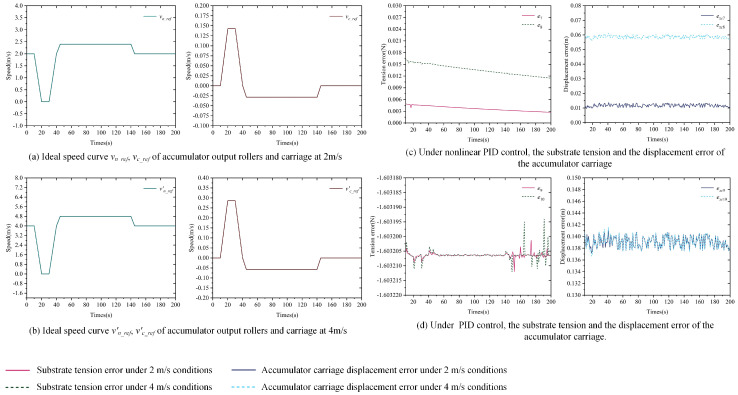
Substrate tension error and accumulator carriage displacement error at 2 m/s and 4 m/s operating speeds using non-linear PID and PID controllers.

**Figure 10 polymers-16-03479-f010:**
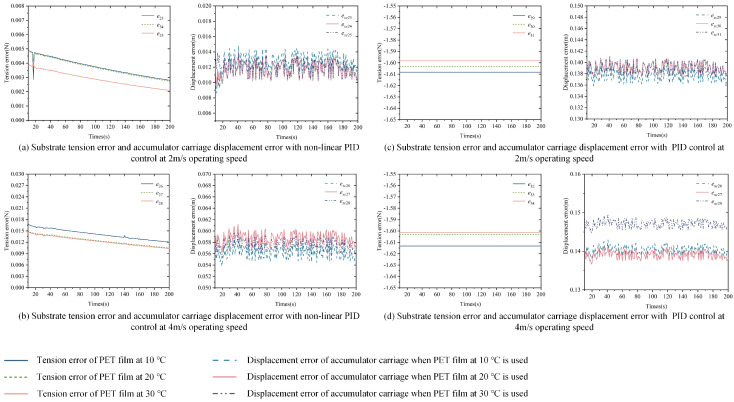
Substrate tension errors and accumulator carriage displacement errors in the control of PET films at different temperatures using nonlinear PID and PID controllers.

**Figure 11 polymers-16-03479-f011:**
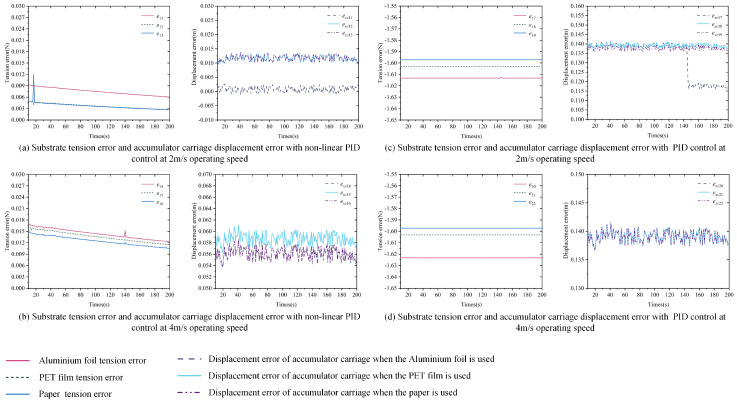
Substrate tension error and accumulator carriage displacement error when controlling different substrates at the same temperature using non-linear PID and PID controllers.

**Table 1 polymers-16-03479-t001:** Parameter meaning table.

Parameter	Mean	Unit
*L*	Length of the segment *i*th substrate	m
*v_i_*	Linear speed of the *i*th roller	m/s
*T_i_*	Tension of the segment *i*th substrate	N
*E*	Substrate Young’s modulus of elasticity	Pa
*A*	Substrate cross-sectional area	m^2^

**Table 2 polymers-16-03479-t002:** Parameter meaning table.

Parameter	Mean	Unit
*f_m_*_0_, *f_mn_*	Bearing friction coefficient	N·m·s
*J*	Roller moment of inertia	kg·m^2^
*J_M_*_0_, *J_Mn_*	Motor output shaft moment of inertia	kg·m^2^
*R_M_*_0_, *R_Mn_*	Motor output shaft radius	m
*u_M_*_0_, *u_M_*_n_	Motor output speed	m/s
*u* * _c_ *	Ball screw thrust	N
*n*	Substrate spans	
*M_c_*	Carriage weight	kg
*g*	Gravitational acceleration	N/kg
*m*	Dynamic friction factor	
$\delta_{1}$, $\delta_{2}$, $\delta_{2}$	Disturbance factor	N
*R*	Roller radius	m

**Table 3 polymers-16-03479-t003:** Parameter meaning table.

Parameter	Mean	Unit
*L_s_*	Ball screw lead	m
*J_Ms_*	Motor output shaft moment of inertia	kg·m^2^
*R_Ms_*	Motor output axis radius	m
$\eta_{3}$	Ball screw transmission efficiency	
$\eta_{1}$, $\eta_{2}$	Gear transmission efficiency	
*N*_1_, *N*_2_	Gear ratio	

**Table 4 polymers-16-03479-t004:** Parameter value table.

Parameters	Value	Parameters	Value
*A*	$1\times 10^{-5}$	*E*	$4.89\times 10^{9}$
*g*	9.8	$M_{c}$	7310
*n*	10	*m*	0.05
*J*	0.2314	*R*	0.2
$R_{M0}$, $R_{Mn}$, $R_{Ms}$	0.03	$f_{m0}$, $f_{mn}$	$2.25\times 10^{-3}$
*J_M_*_0_, *J_Mn_*, *J_Ms_*	0.009	$\eta_{3}$	0.95
$\eta_{1}$, $\eta_{2}$	0.92	*N* _2_	1
*N* _1_	3	*L_s_*	0.01

**Table 5 polymers-16-03479-t005:** Parameter value table.

	TD_1_		TD_1_		NLESF			PID			PID1, PID2, PID3		
Parameters	*r* _0_	*h* _0_	*r* _1_	*h* _1_	*c*	*r* _2_	*h* _2_	*k_p_*	*k_i_*	*k_d_*	*k_p_*	*k_i_*	*k_d_*
Value	2	2	1	10	1.5	2	2	160	100	0.1	900	1000	0.0001

**Table 6 polymers-16-03479-t006:** Young’s modulus and cross-sectional area of the substrates.

Substrate	Temp (°C)	*E*	*A*
PET	10	$5.20\times 10^{9}$	$1\times 10^{-5}$
PET	20	$4.89\times 10^{9}$	$1\times 10^{-5}$
PET	30	$4.59\times 10^{9}$	$1\times 10^{-5}$
Paper	20	$3.80\times 10^{9}$	$1\times 10^{-5}$
Aluminum foil	20	$7.00\times 10^{10}$	$1\times 10^{-5}$

## Data Availability

The original contributions presented in this study are included in the article. Further inquiries can be directed to the corresponding author.

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
