# Peer review of "Modeling of Accumulator in Roll-to-Roll Coating Equipment and Tension Control with Nonlinear PID"

_polymers, 2024, doi:10.3390/polym16243479_

Round 1

Reviewer 1 Report

Comments and Suggestions for Authors

The paper addresses a critical issue in roll-to-roll manufacturing: ensuring high-precision tension control for ultra-thin substrates. It demonstrates thorough modeling by establishing detailed dynamic and coupling models for tension errors, input/output rollers, and carriage dynamics, offering a comprehensive understanding of the system. Another highlight is the innovative use of nonlinear PID controllers as a modern control strategy, showcasing significant performance improvements over classical PID controllers. The simulations using MATLAB/Simulink provide strong evidence to validate the proposed models and control strategies.

Relying solely on simulations without experimental validation raises concerns about the findings' practical applicability. While the nonlinear PID controller exhibits superior performance, the paper must address the real-world challenges associated with its implementation, such as hardware constraints or cost-effectiveness. The comparative analysis could be strengthened by including detailed evaluations against other advanced controllers, such as fuzzy logic or model predictive control, to position the nonlinear PID controller within a broader context of available solutions. Consider the following points to update the paper.

·       What are the practical constraints of implementing nonlinear PID controllers in real-world roll-to-roll systems?

·       Can the models be validated with experimental data or field applications for better reliability?

·       How does the proposed method perform with various substrate materials with different mechanical properties?

·       Are there computational limitations in real-time applications due to the complexity of nonlinear PID controllers?

·       How does the system respond to external disturbances, such as temperature changes or substrate inconsistencies?

·       How does varying operational speeds beyond 1 m/s and 3 m/s affect the controller's performance?

·       What other advanced control strategies (adaptive or neural network-based methods) offer comparable or better results?

·       Are there safety concerns or limitations during sudden tension fluctuations caused by actuator constraints?

·       How does the proposed system handle long-term operational stability and wear of components?

·       Can the nonlinear PID approach be extended to multi-span tension control systems in more complex configurations?

·       What are the energy efficiency implications of using nonlinear PID controllers compared to classical methods?

·       How is the scalability of the proposed approach ensured for larger or more industrial-scale systems?

Comments on the Quality of English Language

The quality of the English language is good, with clear communication of technical concepts.

Author Response

Comments 1: What are the practical constraints of implementing nonlinear PID controllers in real-world roll-to-roll systems?

Response 1: Thank you for making this important point. The practical limitations of using nonlinear PID controllers in real reel-to-reel systems we have considered the following points:

Real-Time Computation: Nonlinear controllers generally require more computational power than linear ones. In roll-to-roll systems, where real-time processing is crucial, it's important to ensure that the controller runs efficiently without delays or excessive computational demands.

Tuning Difficulty: While nonlinear PID controllers can offer better performance, tuning them in practice can be difficult. The trial-and-error or adaptive tuning methods needed may not be practical in fast-paced manufacturing environments where quick setup and frequent product changes are required.

Implementation Costs: The complexity of nonlinear PID controllers often means higher costs for hardware, sensors, and maintenance.

Despite these challenges, we believe that with careful design, simulation, and implementation, nonlinear PID controllers could greatly improve the performance of roll-to-roll systems.

Comments 2: Can the models be validated with experimental data or field applications for better reliability?

Response 2: Thank you for your insightful comments on model validation. We agree that validating models using experimental data or through field applications is critical to ensure their reliability and robustness under real-world conditions.

In our current work, we have focused on developing and testing models in controlled simulation environments to demonstrate their theoretical validity. However, we recognise the need for validation through experimental data to confirm its performance in real roll-to-roll systems.

As part of our future work, we plan to conduct experimental validation on prototypes or in collaboration with industrial partners to collect real-world data to assess the accuracy and reliability of the model.

Comments 3: How does the proposed method perform with various substrate materials with different mechanical properties?

Response 3: Yes, robustness verification experiments for the proposed method are necessary. Therefore, we have made revisions to the resubmitted paper. We have added robustness experiments, focusing on simulations under different operating speeds, various substrate materials, and different experimental temperatures. The results show that the proposed method exhibits good robustness. The specific modifications are in lines 364 to 407 on page 13 of the article.

Comments 4: Are there computational limitations in real-time applications due to the complexity of nonlinear PID controllers?

Response 4: Thank you for your thoughtful comments regarding the potential computational limitations of using nonlinear PID controllers in real-time applications.

The complexity of nonlinear PID controllers can pose a challenge to computational efficiency, especially in applications that have stringent latency requirements or limited hardware resources, where the complexity of nonlinear controllers can indeed pose a limitation. As part of our future work, we plan to explore alternative approaches such as adaptive control strategies, model simplification or hardware acceleration to improve the real-time applicability of the proposed approach.

Comments 5: How does the system respond to external disturbances, such as temperature changes or substrate inconsistencies?

Response 5: Thank you for your comments, when the temperature or substrate changes, this puts a high demand on the performance of the controller, therefore, we added robustness experiments to verify the robustness of the proposed controller.

We have added robustness experiments, focusing on simulations under different operating speeds, various substrate materials, and different experimental temperatures. The results show that the proposed method exhibits good robustness. The specific modifications are in lines 364 to 407 on page 13 of the article.

Comments 6: How does varying operational speeds beyond 1 m/s and 3 m/s affect the controller's performance?

Response 6: Thank you for your comments, this puts a high demand on the performance of the controller when the operating speed of the system is constantly changing, therefore, we added robustness experiments to verify the robustness of the proposed controller.

We have added robustness experiments, focusing on simulations under different operating speeds, various substrate materials, and different experimental temperatures. The results show that the proposed method exhibits good robustness. The specific modifications are in lines 364 to 407 on page 13 of the article.

Comments 7: What other advanced control strategies (adaptive or neural network-based methods) offer comparable or better results?

Response 7: Thank you for your insightful comments. Indeed, advanced control strategies, such as adaptive control and neural network-based approaches, are powerful alternatives that can provide comparable or even superior performance in some cases. In our study, we focus on the proposed controller because of its relative simplicity and robustness.

At present, our team has done a lot of research in this area. For example, in the field of roll-to-roll register control, an improved parameter self-tuning controller For example, in the field of roll-to-roll register control, an improved parameter self-tuning controller combining RBF neural network and ADRC controller is proposed. controller incorporating genetic algorithm and ADRC controller is proposed. In my subsequent research I will gradually verify the control effect of these improved parameter self-tuning controllers in the area of accumulator tension error.

Comments 8: Are there safety concerns or limitations during sudden tension fluctuations caused by actuator constraints?

Response 8: Thank you for raising the safety concerns. In our system, actuator constraints may lead to the following safety issues: On one hand, rapid changes in tension can overload the actuator, potentially exceeding its maximum load and causing damage or failure. On the other hand, sudden tension fluctuations can lead to system instability, triggering oscillations or instability, which may result in inaccurate control and even structural damage.

To prevent such situations in practice, we have designed the actuator's speed profile to be as smooth as possible, avoiding sudden inflection points. Additionally, when the actuator is under constraint, the controller reduces the response speed to ensure the system stabilizes gradually.

Comments 9: How does the proposed system handle long-term operational stability and wear of components?

Response 9: Thank you for your valuable feedback! In response to the concerns about long-term operational stability and component wear, we've simplified our solutions to ensure they are easy to implement and operate:

1. Long-term Operational Stability

To maintain system stability, we have incorporated simple monitoring mechanisms. These monitoring tools continuously check the status of key components (such as motors, sensors, etc.) to ensure that any issues are detected early during operation. If an abnormality is detected in a component, the system will automatically notify for maintenance or replacement, preventing potential failures.

2. Component Wear

To address component wear, we have selected durable materials and designs to extend the lifespan of key parts. We have also added wear monitoring capabilities that regularly check the condition of critical components. Simple sensors and periodic checks allow us to detect wear issues early and take necessary actions, avoiding performance degradation due to excessive wear.

In summary, our design focuses on simplicity and ease of implementation, using real-time monitoring and simple maintenance alerts to ensure long-term operational stability and the health of components.

Comments 10: Can the nonlinear PID approach be extended to multi-span tension control systems in more complex configurations?

Response 10: Thank you for your valuable question. Regarding whether the nonlinear PID method can be extended to more complex multi-span tension control systems, we provide the following response:

1. Scalability of Nonlinear PID

Nonlinear PID controllers can handle complex systems by addressing nonlinearities and dynamic changes. In multi-span tension control systems, this method shows potential as it can effectively manage the interactions between multiple tension points. By dynamically adjusting parameters, nonlinear PID can enhance the system's stability and control precision.

2. Challenges in Multi-Span Tension Control Systems

Multi-span tension systems are more complex than single-span systems because they involve the coordination of multiple tension points, and the interactions between spans are significant. Therefore, nonlinear PID control needs to be specially designed and may require optimization algorithms to mitigate the interference between multiple tension points.

3. Existing Research and Applications

Some existing research and practical applications have shown that nonlinear PID methods are effective in multi-span tension control systems, particularly in industries like textile and rolling. By properly adjusting the control strategies, better tension control can be achieved.

In summary, nonlinear PID has the potential to be applied to complex multi-span tension control systems, but it requires adjustments and optimizations based on specific circumstances.

Comments 11: What are the energy efficiency implications of using nonlinear PID controllers compared to classical methods?

Response 11: Thank you for raising this important question. In response, we offer the following explanation. Nonlinear PID controllers possess adaptive capabilities that allow them to better handle system nonlinearities and dynamic variations. In some scenarios, they can achieve more precise control, minimizing excessive compensation and unnecessary adjustments, which ultimately leads to energy savings. However, it is important to note that the increased complexity and computational demands of nonlinear PID controllers must also be considered. The actual improvement in energy efficiency depends on both the specific characteristics of the system and the degree of optimization applied to the nonlinear PID controller. While more accurate control can result in significant energy savings in some systems, the added computational complexity may not always yield a noticeable improvement in energy efficiency in other cases.

Comments 12: How is the scalability of the proposed approach ensured for larger or more industrial-scale systems?

Response 12: Thank you for the valuable comments. In response to the concern about the scalability of the nonlinear PID controller in large-scale or industrial systems, we plan to make optimizations in the following areas:

1. Computational Efficiency Optimization

Although the nonlinear PID controller requires more computation compared to the traditional PID controller, we have significantly improved its computational efficiency by introducing optimized algorithms. This allows the controller to meet the real-time computational demands of larger systems, thus avoiding excessive computational load.

2. Distributed Control

For particularly large systems, we propose adopting a distributed control strategy. By distributing the controllers across different parts of the system and running them independently, we can effectively alleviate the computational pressure on individual controllers, significantly improving the scalability and stability of the system.

3. Model-based Tuning

We also employ a model-based tuning method to ensure that the controller can be precisely adjusted according to the specific characteristics of the system. This allows the nonlinear PID controller to maintain excellent control performance across different system scales.

4 Robustness and Stability

The nonlinear PID controller demonstrates strong robustness in the face of complex disturbances and uncertainties in large-scale systems. Through precise tuning mechanisms, the system can maintain stable operation and deliver consistent control performance in dynamic environments.

In conclusion, we are confident that, with the above optimizations, the nonlinear PID controller can be effectively scaled to larger systems and achieve excellent control performance in industrial applications.

Reviewer 2 Report

Comments and Suggestions for Authors

The manuscript presents a tension control strategy based on PID and nonlinear PID control algorithms for the accumulator system of a roll-to-roll manufacturing process. The paper is written pretty well. However, there are several sections that need to be improved.

1.      The abstract is too long and includes too detailed information. The abstract should be concise while mentioning the importance of the studied problem and the proposed solution with its originality and performance.

2.      On line 182 it is stated the u_c in eq 5 refers to F_s, but eq 5 does not include u_c.

3.      The subsection 3.1 introduces the equations used to implement the nonlinear PID without a thorough explanation on how these equations were derived, or any reference to the work describing this development. Moreover, the translation from the variables of the nonlinear PID (y, y_ref, u) to the process variables is not provided.

4.      The nonlinear PID and the classic PID involve a significant number of parameters. How were these parameters tuned?

5.      The control diagram in Figure 8a indicates 3 PID controllers, while Table 5 shows only one set of parameters. What about the parameters of the other 2 PID controllers?

6.      On line 308 the reference should be Figure 9(a) and 9(b).

7.      The comparison between the nonlinear PID and the classical PID is made based on the results shown in Figures 9 and 10, which illustrate only the process output. Figures showing all important variables (references, inputs, outputs) in the control loop should be included for a proper evaluation.

Author Response

Comments 1: The abstract is too long and includes too detailed information. The abstract should be concise while mentioning the importance of the studied problem and the proposed solution with its originality and performance.

Response 1: Thank you for pointing this out. We agree with this comment. Therefore, we've made the changes in the resubmitted article, which are on lines 1 to 18 on the first page of the article.

Comments 2: On line 182 it is stated the uc in eq 5 refers to Fs, but eq 5 does not include uc.

Response 2: Thank you for pointing this out. We agree with this comment. Therefore, we have corrected the error in equation (5) on page 5 of the resubmitted article and added an explanation for the variable uc in Table 2.

Comments 3: The subsection 3.1 introduces the equations used to implement the nonlinear PID without a thorough explanation on how these equations were derived, or any reference to the work describing this development. Moreover, the translation from the variables of the nonlinear PID (y, y_ref, u) to the process variables is not provided.

Response 3: We agree with you that we are missing some necessary explanations of non-linear PID controllers. Therefore, we have made the necessary changes in our resubmitted article, mainly by citing some relevant references and adding explanations of relevant parameters in nonlinear PID controllers. The changes are shown on page 10, lines 277 to 304 of the article.

Comments 4: The nonlinear PID and the classic PID involve a significant number of parameters. How were these parameters tuned?

Response 4: Thank you for raising this important question regarding the parameter tuning of both nonlinear and classic PID controllers. In our study, the parameters for both controllers were tuned using a systematic approach to ensure optimal performance.

For the classic PID controller, we employed the widely used Ziegler-Nichols method for initial tuning, followed by fine-tuning based on simulation results. The proportional, integral, and derivative gains were adjusted iteratively to achieve a balance between response speed and stability, minimizing both overshoot and steady-state error.

For the nonlinear PID controller, parameter tuning was more complex due to the nonlinear nature of the control algorithm. We initially set the parameters based on theoretical guidelines and then used a trial-and-error approach to fine-tune the gains for better performance under varying operating conditions.

Comments 5: The control diagram in Figure 8a indicates 3 PID controllers, while Table 5 shows only one set of parameters. What about the parameters of the other 2 PID controllers?

Response 5: Thank you for pointing this out, and in Table 5 on page 12 of the new submission, we have added three additional parameter values for the PID controller.

Comments 6: On line 308 the reference should be Figure 9(a) and 9(b).

Response 6: Thank you for pointing this out and making the appropriate change on page 12, line 322 of the newly submitted article.

Comments 7: The comparison between the nonlinear PID and the classical PID is made based on the results shown in Figures 9 and 10, which illustrate only the process output. Figures showing all important variables (references, inputs, outputs) in the control loop should be included for a proper evaluation.

Response 7: Thank you for your valuable feedback. We have made targeted revisions in the resubmitted manuscript, which can be found on pages 8, lines 208 to 218. The reason for the revision is as follows: Based on our analysis of the relationship between the displacement error of the accumulator carriage and the tension error of the base material, we also observed the speed errors between the input and output rollers of the accumulator in the absence of a controller. The results show that when the output speed of the accumulator’s input and output rollers closely follows the ideal speed curve, the displacement error of the accumulator carriage is the main source of the tension error in the base material. Therefore, based on this simulation result, when comparing the tension error control performance of the nonlinear PID controller with that of the conventional PID controller, we focused on observing the variations in tension error and carriage displacement error.

Round 2

Reviewer 1 Report

Comments and Suggestions for Authors

The authors revised as per suggestions.

Reviewer 2 Report

Comments and Suggestions for Authors

The quality of the figures showing the simulation results should be improved. They are quite blurry and the text size is too small.